# Reduction spheroids preserve a uranium isotope record of the ancient deep continental biosphere

Sean McMahon[1,2], Ashleigh v.S. Hood[1,3], John Parnell[4] & Stephen Bowden[4]

Life on Earth extends to several kilometres below the land surface and seafloor. This deep biosphere is second only to plants in its total biomass, is metabolically active and diverse, and is likely to have played critical roles over geological time in the evolution of microbial diversity, diagenetic processes and biogeochemical cycles. However, these roles are obscured by a paucity of fossil and geochemical evidence. Here we apply the recently developed uranium-isotope proxy for biological uranium reduction to reduction spheroids in continental rocks (red beds). Although these common palaeo-redox features have previously been suggested to reflect deep bacterial activity, unequivocal evidence for biogenicity has been lacking. Our analyses reveal that the uranium present in reduction spheroids is isotopically heavy, which is most parsimoniously explained as a signal of ancient bacterial uranium reduction, revealing a compelling record of Earth's deep biosphere.

[1] Department of Geology and Geophysics, Yale University, P.O. Box 208109, New Haven, CT 06520-8109, USA. [2] UK Centre for Astrobiology, School of Physics of Astronomy, University of Edinburgh, James Clerk Maxwell Building, Edinburgh EH9 3FD, UK. [3] School of Earth Sciences, University of Melbourne, Parkville, VIC 3010, Australia. [4] School of Geosciences, University of Aberdeen, Aberdeen AB24 3UE, UK. Correspondence and requests for materials should be addressed to S.M. (email: sean.mcmahon@ed.ac.uk)

The subsurface represents a vast habitat containing up to a fifth of Earth's current total biomass, including diverse bacteria, archaea, and fungi[1,2]. In recent decades, accelerating exploration of the deep biosphere has revealed new microbial groups, new ecological niches, and new modes of microbe–mineral interaction[3–5]. However, a full understanding of the limits of the deep biosphere, its contribution to total planetary biomass, and its biogeochemical significance requires the detection and analysis of robust deep biosignatures in the rock record. Such biosignatures could also refine search images for past or present life on Mars, whose surface has long been uninhabitable but may conceal a warm, wet interior[6].

It is difficult to confirm that possible traces of the ancient deep biosphere are truly biogenic, post-burial in origin, and pre-modern. Few reported cellular or molecular fossils pass all three tests unequivocally, and most candidates represent subseafloor environments[7–10]. Microbially mediated diagenetic phenomena on palaeo-redox fronts offer a potentially powerful alternative record that could extend to continental settings[11,12]. Of long-standing interest in this connection are reduction spheroids, which are very common mm–dm-scale bleached spots found most commonly (but not exclusively) in Proterozoic–Phanerozoic red beds, i.e., sedimentary rocks deposited in oxidising terrestrial environments and rich in early diagenetic haematite[13,14]. The bleached colour reflects localised Fe(III) reduction and loss[15]. Many examples contain small, dark, central "cores" where redox-sensitive elements including uranium, vanadium and nickel are highly concentrated and organic matter may also be present[16,17].

The mechanisms that produce reduction spheroids in the subsurface have hitherto been unclear. They are sometimes attributed to the oxidation of organic-rich cores, but the cores are usually darkened not by organic matter but by opaque metalliferous minerals. Although organic carbon is present in some reduction spheroid cores and may have stimulated their formation, organic carbon present prior to reduction is typically too scarce to have reduced the surrounding halos[16,17]. It has thus been proposed instead that spheroids form around localised chemolithotrophic microbial populations, which would catalyse the oxidation of mobile reductants supplied through groundwater[13,18–20]. These reductants could include $H_2$ derived radiolytically from porewater, which could establish a positive-feedback mechanism for spheroid growth following uranium precipitation[20]. Confirmation of a biotic mode of origin would distinguish these common geological features as perhaps the most widely accessible, recognisable and distinctive traces of the ancient deep biosphere. It would also add weight to previous suggestions that reduction spheroids could be a target for astrobiological sampling on Mars, where iron reduction is regarded as a plausible metabolic strategy for past or present life[21–23].

Direct evidence for the biogenicity of reduction spheroids has hitherto been lacking or equivocal. Authigenic pyrite present in some spheroids has a sulphur isotope composition consistent with but not diagnostic of a bacterial origin[21]; however, most spheroids lack pyrite altogether[19]. Molecular biomarkers can be extracted from the organic matter commonly associated with reduction spheroids (e.g., Supplementary Figure 1), but are likely to pre-date the origin of the spheroids themselves, so cannot shed light on their biogenicity. This organic matter is also damaged and isotopically modified by exposure to ionising radiation commonly emitted by uranium in reduction spheroids[17,19,24]. Hence, determination of reduction spheroid biogenicity is non-trivial and necessitates the analysis of authigenic phases demonstrably associated with spheroid formation.

Here, we focus on a new low-temperature palaeo-redox proxy, the isotopic composition of uranium ($^{238}U/^{235}U$: $\delta^{238}U$, in standard delta notation, relative to the CRM-112a standard; Eq. (1)). Uranium enrichment appears to be a universal feature of reduction spheroids, occurring both in the cores and in the halos as a result of the highly localised reduction of soluble U(VI) to insoluble U(IV) (ref. [19]). Thus, uranium phases (both mineralised and non-mineralised) in reduction spheroids and analogous low-temperature redox-front uranium deposits have been shown[12,19] to contain predominantly U(IV). Uranium reduction can occur via many pathways[18,25], both abiotic (coupled to the oxidation of various aqueous, mineral, and organic species) and biotic (i.e., enzymatic catalysis by chemolithotrophic microorganisms capable of facultatively utilising U(VI) as an electron acceptor, including iron- and sulphate-reducers). The uranium isotope system is controlled by low-temperature redox reactions that significantly fractionate the uranium isotope composition preserved in environmental samples away from a crustal (high-T) average $\delta^{238}U$ of $-0.29 \pm 0.03$‰, often concentrating the heavier isotope in the reduced product[26,27].

Several experimental studies have shown that bacterially reduced and precipitated uranium is isotopically heavier (i.e., records higher $\delta^{238}U$) than the dissolved precursor phase[28–30]. Bhattacharyya et al.[12] recently determined a bacterial origin for isotopically heavy authigenic uranium phases in roll front ore deposits, which resemble reduction spheroids inasmuch as they are mineralised paleo-redox fronts formed at low temperatures in subsurface aquifers. Field studies confirm that modern groundwaters inoculated with metal reducing bacteria become isotopically lighter in uranium as the heavier isotope is preferentially precipitated[31,32]. Experimental studies so far have shown that, by contrast, abiotically reduced uranium either remains unfractionated or is isotopically lighter, regardless of the reductant responsible[29,30,33,34]. Consequently, the U-isotopic composition ($\delta^{238}U$) of reduced uranium phases in nature is emerging as a new and potentially powerful proxy for their mode of origin[12,30,31] (Supplementary Note 1).

Here, we report $\delta^{238}U$ analyses of the dark cores, bleached halos, and surrounding matrix of reduction spheroids collected from continental red beds in outcrop, primarily at Dingwall in northern Scotland and Budleigh Salterton in southwest England, sites where spheroids are both especially uraniferous, and can be linked to unusually well constrained formation depths from their geological context. Spheroids from other localities of diverse ages were analysed for comparison, as were uraniferous hydrothermal veins expected to yield near-crustal $\delta^{238}U$ values reflecting their high-temperature origin[30,34]. We find that reduction spheroids are enriched towards their cores in uranium characterised by high $\delta^{238}U$ values. This result is most parsimoniously explained as a signal of ancient bacterial U(VI) reduction, implying that the spheroids themselves are most likely bacterial in origin.

## Results and discussion

**Uranium isotope values.** The cores of reduction spheroids have uniformly higher uranium concentrations and heavier uranium isotope compositions ($\delta^{238}U$) compared to the host rock in all samples (Supplementary Table 1; Fig. 1). All hydrothermal vein samples and most of the red-bed matrix from the reduction-spheroid localities yielded $\delta^{238}U$ values near the average crustal value of $-0.29 \pm 0.03$‰ (ref. [27]). In most reduction spheroids, both uranium concentration and $\delta^{238}U$ increased from the matrix through the halo into the core. At Budleigh Salterton, the large size of the spheroids made it possible to discriminate between isotopically heavier black inner cores (mean +0.78‰; $n = 4$) and isotopically less heavy dark grey core margins (+0.07‰; $n = 2$), as well as greenish outer halos (−0.24‰; $n = 4$). Similarly, the spheroid cores from Dingwall yielded much heavier values of $\delta^{238}U$ (mean +0.45‰; $n = 6$) than the halos (+0.04‰; $n = 4$),

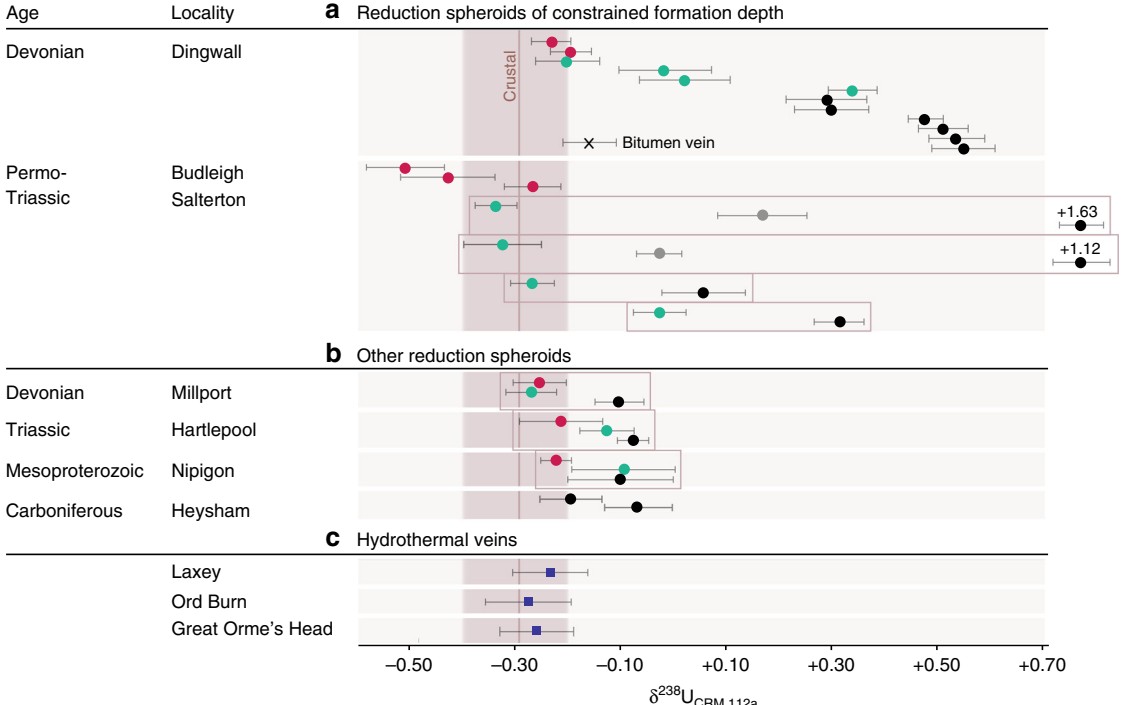

**Fig. 1** Results of uranium isotope analysis. Error bars represent two standard errors. $\delta^{238}U$ values are shown from reduction spheroid innermost cores (black), core margins (grey), bleached halos (cyan), and surrounding red-bed matrix (magenta). Boxes indicate physically contiguous samples. **a** Results from reduction spheroids of constrained formation depth. **b** Results from other reduction spheroids. **c** Results from three hydrothermal veins

the matrix ($-0.21‰$; $n = 2$), and a nearby bitumen vein ($-0.16‰$; $n = 1$). Spheroid cores from the other localities, where palaeodepth was less well constrained, also recorded values heavier than the crustal range, and all were heavier than their respective matrices by at least 0.10‰ (Supplementary Table 1). Uranium activity ratios ($234U/238U$) are given in Supplementary Table 1 and Supplementary Figure 2.

**Reduction spheroid biogenicity**. The reduction spheroids analysed here are enriched in uranium and show increasingly heavy isotopic compositions ($\delta^{238}U$) towards their reduced cores. In some cases, the matrix to core interval of reduction spheroids expresses U isotope variation approximating almost the entire natural range of low-temperature systems on Earth[26,35]. Field-based, experimental and geological studies to-date strongly suggest that these high $\delta^{238}U$ values are best explained by bacterial uranium reduction and precipitation within the spheroids[12,29–34]. In particular, given the environmental similarity between reduction spheroids and roll-front ore deposits, our interpretation receives support from the recent measurement of isotopically heavy biogenic uranium phases associated with the latter[12].

Since Fe(III) and U(VI) reduction are carried out by the same groups of microorganisms using the same reductants, and occur coextensively and concurrently in modern aquifers[36], our results strongly imply that the reduction and dissolution of ferric iron responsible for the presence of the bleached spheroids themselves was also bacterially mediated. We infer that reduction spheroids, which are both spatially and temporally widespread, represent an important record of the geological history of the deep biosphere, which was potentially Earth's largest reservoir of biomass prior to the proliferation of land plants[37].

**A record of the ancient deep biosphere**. The deep biosphere conventionally extends from ~metres depth to several kilometres[4,38,39]. There is clear evidence that many—perhaps most—

reduction spheroids form at the deeper end of this range. In brief: first, halos are commonly spherical, whereas shallow non-nodular features would be flattened by compaction; second, radiometric ages of authigenic minerals concentrated within some spheroids are $>10^7$ years younger than the host rock[16,40]; third, some spheroids occur in haematite-stained igneous basement, hundreds of metres below the uppermost basement[14]; fourth, at many localities, the distribution of spheroids was clearly influenced by pre-existing faults, fractures, cataclastic zones and cleavages younger than the host rock[14,41,42]. Our findings evince a clear signal of bacterial uranium reduction in spheroids demonstrably formed at multi-km depth, including one locality (Dingwall) where they appear to be related to the early-stage biodegradation of hydrocarbons, and a weaker but consistent signal at all other localities. We conclude that reduction spheroids represent an important and widespread archive of the deep continental biosphere, present through much of Earth's geological record. This finding lends weight to the suggestion that reduction spheroids be targeted for analysis and sample return were they to be discovered on Mars[22].

## Methods
**Sample localities**. The Dingwall spheroids are hosted by red mudstones of the middle Devonian Millbuie Sandstone Group, which forms part of a thick continental succession (the Old Red Sandstone). As previously described by ref. [24], the cores are black, spherical nodules a few mm across, composed of solid hydrocarbons with uranium present as microscopic inclusions of uraninite and other minerals (impure xenotime and possibly brannerite). These cores occupy green-grey non-nodular halos that can extend for several centimetres, and occur through a stratigraphic thickness of 10 m (Fig. 2). The post-compaction origin of the Dingwall spheroids is confirmed by (1) the spherical shape of the cores; (2) the lack of compaction drapes over them; and, (3) the presence of solid hydrocarbon residues (bitumen) within the cores that clearly derive from source rocks in the underlying kilometre of stratigraphy[24], which must have been deeply buried in order to reach thermal maturity and generate hydrocarbons (i.e., about 3 km assuming a normal geothermal gradient). Migration occurred while the succession was still deeply buried, as demonstrated by the presence of bitumen-bearing quartz veins through a stratigraphic thickness of ~10 m in the conglomerate directly

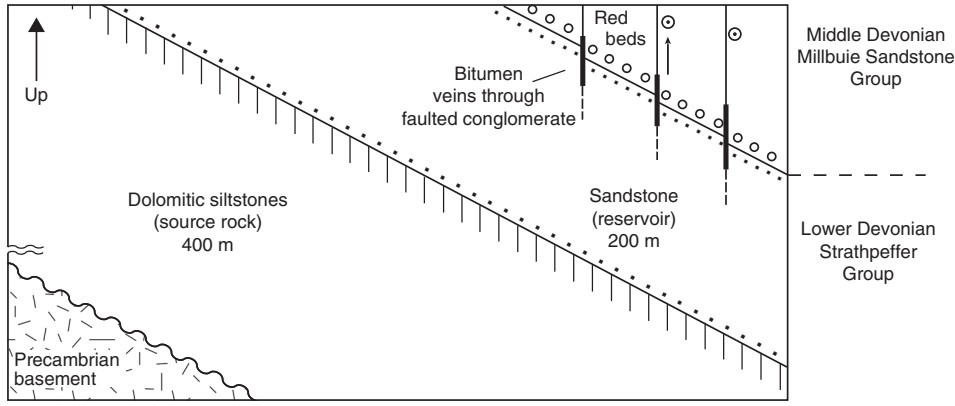

**Fig. 2** Schematic view of geological context of Dingwall samples. Key to symbols: stipples = igneous basement; parallel lines = mudrock; dots = sandstone; open circles = conglomerate; bulls' eyes = reduction spheroids. Bitumen is present at the cores of the reduction spheroids and in the local fractures. Adapted from ref. [24]

underlying the spheroid-hosting mudstone[24]. These veins, which yield fluid-inclusion temperatures close to 100 °C (ref. [43])—equivalent to ~3–4 km depth assuming a normal geothermal gradient—occur within 5 m of the spheroids themselves. We, therefore, infer that the Dingwall spheroids formed at depths of several kilometres and may be genetically related to hydrocarbon migration and (bio)degradation.

The reduction spheroids at Budleigh Salterton are hosted by red mudstone in the latest Permian Littleham Mudstone Formation, a ~200-m thick unit within the New Red Sandstone Supergroup, a laterally and stratigraphically extensive Permo-Triassic continental succession[44]. The specimens analysed here are spherical pale green nodules with diffuse black centres; uranium is present as fine-grained coffinite and is not associated with organic matter[45,46]. Cross-cutting relationships described by ref. [38] show that these spheroids formed penecontemporaneously with sheet-like copper nodules, which themselves replaced an earlier generation of crack-seal calcite veins generated by overpressure during compaction dewatering. These relationships suggest the Budleigh Salterton reduction spheroids formed relatively early, at depths of up to around 1–2 km (ref. [46]).

In addition to Budleigh Salterton and Dingwall, reduction spheroids were collected from the field at four other localities: (1) the Mesoproterozoic (~1.4 Ga) red beds of the Sibley Group of Ontario, sampled at a road cut near Nipigon[47]; (2) Devonian red sandstone at Millport, Great Cumbrae, Scotland; (3) Carboniferous white sandstone at Heysham, Lancaster, England[24]; (4) Triassic red siltstone at Hartlepool, Co. Durham, England. The sample from Ord Burn was collected from a hydrothermal vein in Caledonian granite, Sutherlandshire, Scotland[48]. The sample from Great Orme's Head is a copper ore deposit in Carboniferous limestone, North Wales[49]. The sample from Laxey is a hydrothermal vein-hosted lead–zinc ore deposit in Lower Palaeozoic slates above Caledonian granite, Isle of Man[50].

**Sample preparation.** Samples were cut, cleaned and crushed in an agate mill, or micro-drilled with a tungsten carbide drill bit (previously tested to not contaminate U isotope analysis, and cleaned between samples) to target the specific components of reduction spheroids, host rocks and other samples (Fig. 3). Approximately, 0.1–0.4 g of each powdered and homogenised sample were ashed in a 100 °C oven for 24 h. Samples were digested in a 3:1 mixture of concentrated $HNO_3$ and HF on a hotplate for 24 h. Samples were dried and re-digested in concentrated HCl and $HNO_3$.

**Isotope analyses.** Trace metal concentrations were measured at the Yale Metal Geochemistry Center on a Themo-Finnigan Element XR ICP-MS on splits from each digest. The $^{236}U$–$^{233}U$ double spike was added based on uranium concentrations ($^{238}U/^{236}U$ ~30), prior U purification via ion exchange methods. The spiked samples were dried and taken up in 3N $HNO_3$. The U was then purified using the UTEVA column chemistry method (after ref. [51]; see methods of ref. [39,52]). Purified U was dissolved in 0.75 N $HNO_3$ with 50 ppb concentration. Uranium isotopes were measured at the Yale Metal Geochemistry Center on a Themo-Finnigan Neptune Plus Multi-Collector ICP-MS at low-mass resolution using a Jet sampler cone and a standard skimmer cone. Sample were introduced through an Elemental Scientific μFlow PFA nebuliser at ~50 μL/min via an Elemental Scientific Apex IR. A 50 ppb sample solution yielded 32–40 V of $^{238}U$ signal on a $10^{11}$ Ω amplifier.

Isotopes were measured on Faraday collectors, listed in Table 1. $^{232}Th$ hydride was monitored to have a negligible effect on measurement of $^{233}U$. Measurements consisted of five blocks, each block ten cycles, each cycle 4.19 s. Blank U level was less than 50 pg. External reproducibility was assessed using full protocol duplicates

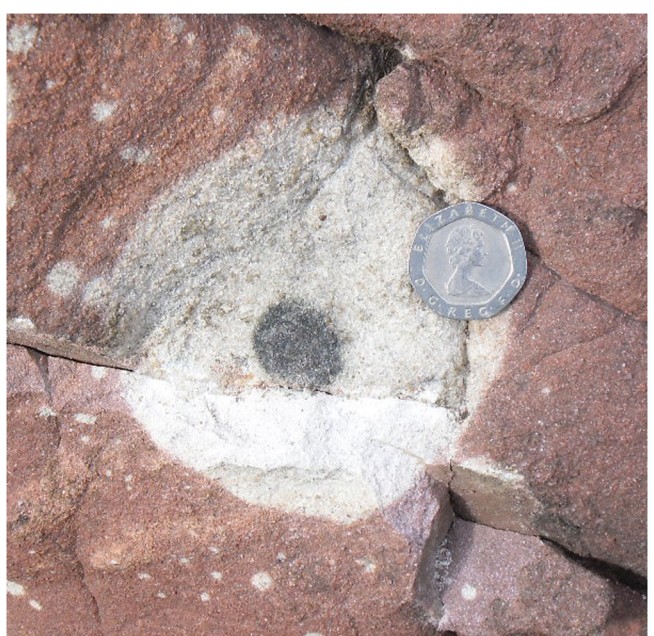

**Fig. 3** Example of a freshly exposed reduction spheroid. This spheroid shows a distinctive dark grey core, a bleached halo and a red-brown matrix flecked with smaller reduction spots. The coin is ~21 mm across. Devonian red sandstone, Millport, Great Cumbrae, Ayrshire, Scotland

**Table 1 Neptune Faraday detector setup**

| Isotope | $^{232}Th$ | $^{233}U$ | $^{234}U$ | $^{235}U$ | $^{236}U$ | $^{238}U$ |
|---|---|---|---|---|---|---|
| Cup | L3 | L2 | L1 | C | H1 | H3 |
| Amplifier | $10^{11}$ | $10^{11}$ | $10^{12}$ | $10^{11}$ | $10^{11}$ | $10^{11}$ |

of the geostandard NOD-A-1, which yielded an average δ238U of −0.52 ± 0.08‰ based on nine repeats (2σ error = 0.12). Duplicate samples agreed within error.

Uranium isotope variations of samples and standards are reported as δ238U$_{CRM 112a}$, which is defined as:

$$\delta^{238}U = \left( \left[ \frac{\left(\frac{^{238}U}{^{235}U}\right)_{sample}}{\left(\frac{^{238}U}{^{235}U}\right)_{CRM-112a}} \right] - 1 \right) \times 1000‰ \qquad (1)$$

**Biomarker analyses.** Quantitative biomarker data (Supplementary Note 2) were obtained by gas chromatography-mass spectrometry (GC–MS) from the spheroids from Dingwall as follows. Core samples were prepared by rinsing twice with

distilled water and again with dichloromethane (DCM), and ultrasonicated with DCM and methanol. All glassware was thoroughly cleaned with a 93:7 mixture of DCM/MeOH. Crushed samples were weighed, recorded, transferred into pre-extracted thimbles, dried with a rotary evaporator, and separated into aliphatic, aromatic and polar fractions via silica column chromatography using hexane, hexane/DCM in the ratio 3:1 and DCM/MeOH, respectively. Prior to GC–MS analysis, an internal standard (5β-Cholane, Agilent Technologies) was added to the saturated fraction before injection into the GC–MS machine, and subsequent biomarker identification. This was done using an Agilent 6890N gas chromatograph fitted with a J&W DB-5 phase 50 m MSD and a quadruple mass spectrometer operating in SIM mode (dwell time 0.1 s per ion and ionisation energy 70 eV). Samples were injected manually using a split/splitless injector operating in splitless mode (purge 40 ml min$^{-1}$ for 2 min). The temperature programme for the GC oven was 80–295 °C, holding at 80 °C for 2 min, rising to 10 °C min$^{-1}$ for 8 min and then 3 °C min$^{-1}$, and finally holding the maximum temperature for 10 min. Data were obtained by comparing with the response of the internal standard.

## Data availability

All data generated during and/or analysed during this study are available from the corresponding author upon reasonable request.

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

## Acknowledgements

S.M. acknowledges the support of the NASA Astrobiology Institute grant NNA13AA90A, Foundations of Complex Life, Evolution, Preservation and Detection on Earth and Beyond, and the European Union's Horizon 2020 Research and Innovation Programme under Marie Skłodowska-Curie grant agreement 747877. Av.S.H. was supported by a NASA Astrobiology Institute Postdoctoral Fellowship and acknowledges the support of Xiangli Wang and Devon Cole for lab assistance. S.M. and Av.S.H. thank Noah Planavsky for technical advice, lab support, and comments on an early draft. J.P. was supported by NERC under grant number NE/L001764/1. The isotope facility at SUERC is supported by NERC. The authors thank the two anonymous referees for constructive criticisms that improved the manuscript.

## Author contributions

S.M. conceived the study and wrote the manuscript. S.M., J.P., and Av.S.H. prepared the samples. Av.S.H. conducted the isotopic analyses and wrote the methods section. J.P. contributed the samples and supporting locality information. S.B. conducted the GC–MS analysis. All authors contributed to the design of the study and edited the manuscript.

## Additional information

**Competing interests:** The authors declare no competing interests.

