## [Peer Review File · Nature Communications]

Reviewers' comments:

Reviewer #1 (Remarks to the Author):

Comments to the authors:

The manuscript entitled "A Uranium Isotope Record of the Deep Continental Biosphere" reports about Uranium isotope fractionation as an indicator for bacterial activity in Uranium spheroids in continental rocks.

The general content is informative and maybe of interest for a broad readership. The method presented seems useful, but needs much more detailed discussion and validation including data about the spheroids themselves and the organic content. So far, the general argumentation for a biotic signature is rather weak and in the current state it won't convince the reader by presenting a half-hearted correlation of the uranium isotope fractionation, the uranium concentration with an undefined organic fraction (%loss by heating?!). Especially if the authors assume a biogenic spheroid formation with bacteria involved in uranium reduction, I strongly recommend checking for bacterial traces i.e. biomarker. There are numerous geomicrobiological studies describing different bacteria involved in uranium reduction such as iron reducing bacteria and sulphate reducing bacteria (Ontiveros-Valencia et al., 2017; Komlos et al., 2008; Ollmann et al., 2006; Wilkins et al., 2005). Biomarker for these bacteria are well described (e.g. Taylor & Parks 1985; Dowling 1986; Vainsthein et al., 1992; Härtner et al., 2005) and the size of the spheroids would allow a sampling and biomarker investigation (core, halo, adjacent sediment) for microbial lipids. Others describe in detail how and where the uranium could interact/be trapped together with the lipids (Alessi et al., 2014) especially phosphate bearing head groups. A convincing example for biosignatures from a deep continental biosphere .i.e. extreme biogenic isotope fractionation values in combination with complementary techniques was published by Drake et al 2015.

Find below some titles and excerpts of the recommended publications which may help to address the requested changes.

- Ontiveros-Valencia et al., 2017 Total electron acceptor loading and composition affect hexavalent uranium reduction and microbial community structure in a membrane biofilm reactor.

"SRB Desulfovibrionales, along with sulfur-oxidizers Thiobacteriales, were important in the biofilm during the initial phase that involved SO₄²⁻ reduction; however, only Desulfovibrionales remained in the biofilm that carried out U(VI) reduction. The main URB appeared to be spore-forming Clostridiales and Natranaerobiales, which thrived at low total electron acceptor loadings. Respiration of HCO₃⁻ by methanogenesis (by Methanobacteriales) was important when the only competing electron acceptor was U(VI), but diminished with addition of BES, the onset of SO₄²⁻ reduction, or major homo-acetogenesis."

- Alessi et al., 2014 The product of microbial uranium reduction includes multiple species with U(IV)-phosphate coordination.

"...we have demonstrated unambiguously that noncrystalline U(IV) is comprised of multiple U(IV) species bonded to phosphate groups. Some species are associated with phosphate functional groups in the microbial biomass as either monomers or polymers while others form through the precipitation of inorganic phosphate polymers that are more difficult to extract using a concentrated bicarbonate solution."

- Drake et al., 2015 Extreme ¹³C depletion of carbonates formed during oxidation of biogenic methane in fractured granite.

"...Here we show that oxidation of biogenic methane in carbon-poor deep groundwater in fractured granitoid rocks has resulted in fracture-wall precipitation of the most extremely ¹³C-depleted carbonates ever reported, δ¹³C down to 125‰ V-PDB. A microbial consortium of sulphate reducers and methane oxidizers has been involved, as revealed by biomarker signatures in the carbonates and S-isotope compositions of co-genetic sulphide...."

- Komlos et al., 2008. Long-term dynamics of uranium reduction/reoxidation under low sulfate conditions.

- Ollmann et al., 2006. Metal binding by bacteria from uranium mining waste piles and its technological applications.

"...Bacillus sphaericus JG-A12 from a uranium mining waste pile in Germany are able to accumulate high amounts of toxic metals such as U, Cu, Pb, Al, and Cd..."

- Wilkins et al 2005 The impact of Fe(III) reducing bacteria on uranium mobility
- Härtner et al., 2005. Occurrence of hopanoid lipids in anaerobic Geobacter species
- Vainshtain et al 1992 Cellular Fatty Acid Composition of Desulfovibrio Species and Its Use in Classification of Sulfate-reducing Bacteria
- Taylor & Parks 1985 Identifying Different Populations of Sulphate-reducing Bacteria within Marine Sediment Systems, Using Fatty Acid Biomarkers.

Reviewer #2 (Remarks to the Author):

Review: 'A uranium isotope record of the deep continental biosphere' by McMahon et al.

The study by McMahon, v. s. Hood and Parnell is looking at $^{238}\text{U}/^{235}\text{U}$ signatures in a range of reduced spheroids in ancient red beds. The spheroids are believed to have been formed at considerable depth and, if a biotic origin can be confirmed it would provide evidence for bacterial activity deep in the continental crust during Earth's earlier history. The study shows heavier U isotope signatures in U rich centres of spheroids, which strongly suggest a biotic origin for reduced and precipitated U within these haloes. I am not an expert on spheroid and deep Earth biological activity, but I find the study exciting from an U isotope perspective, its importance seems clear to me and the manuscript is well written. It should have a wide audience and I would find it suitable for publication in Nature Communication. Below, I have some comments to specific places in the manuscript that would be good to be addressed by the authors.

Specific comments

Line 45-46: 'ancient' twice in sentence, consider rewriting

Line 58: 'and' instead of 'are'

Line 104-106: I would like to see some word of caution here, as it is clear that you can generate heavier U isotopes through the nuclear field shift effect through abiotic U redox changes. This has been shown in a range of studies focussing on U redox exchange processes (e.g. Bigeleisen 1996, Fujii et al. 2007). It is true that the cited studies have perhaps focussed on abiotic U reduction processes that are closer to 'natural' situations, so there may be an argument for the distinction between abiotic vs biotic reduction based on these, however more work on this is needed before it is clear that abiotic U reduction does not lead to fractionation with heavier U in the reduced phase. Something like 'Experimental studies so far have shown that, by contrast, abiotically ...' or similar wording could cover this caveat.

Line 107-109: Following on from above, I am sceptic about the statement that isotopically heavy U being a 'powerful proxy' for 'their mode of origin' (abiotic vs. biotic). Making this into a 'powerful proxy' will need much more work and simplistic statements misrepresent where the field is in regard to such understanding. The more extended discussion on this in the supplementary material does provide some further information, but it still misses some fundamental important observations from the literature. There are plenty of examples of measurements of U isotopes in nature that are significantly heavier than the crustal average, and is not related to biotic U reduction e.g. zircons and other accessory high temperature phases (Hiess et al. 2012). Also, heavier (and lighter) U isotopes are associated with seawater alteration of mafic ocean crust (Andersen et al. 2015, Noordmann et al. 2016), which is uncertain how this relates to biotic or abiotic processes. That said, in this case of studying these reduction spheroids, the similarity with redox-front hosted U deposits are probably of more significant importance. Redox-front U systems

are very likely microbial driven and leads to heavy U isotope signatures in the reduced phase (as discussed earlier in the manuscript and supplementary material). Thus, if similar behaviour is observed in the reduction spheroids, this is likely linked to the same process and therefore it can, in this case, be used to test for if the redox haloes are likely to be biotic in origin. This observation should form the basis for the interpretations here and it would be good if the text reflects this better.

Line 122-152: I am not an expert on spheroids and from reading these descriptions it is not clear to me if the interpretations of their origin at depth in the basement is based on observations done in this study or if it is based on literature work e.g. line 126-135 is all or only some of these observations from Parnell & Eakin, 1987? Although the one author from this study is also on the paper, it would be good to make this clearer and if these descriptions are presented for the first time and if the case, some supplementary material to describe and support these statements in more detail would seem appropriate. Also, all these settings are Phanerozoic (apart from one of the additional samples) so it is not expected to be deep biological activity happening?

Line 176-177: the natural range in $^{238}\text{U}/^{235}\text{U}$ is larger than the 2 permil variability observed here (e.g. see compilation in Hiess et al 2012).

Line 177-179: see comments line 104-109 – ‘strong evidence’, maybe reword.

Line 201-202: two times ‘demonstrate’ in the same sentence consider rewriting.

Reviewer #1 (Remarks to the Author):

Comments to the authors:

The manuscript entitled “A Uranium Isotope Record of the Deep Continental Biosphere” reports about Uranium isotope fractionation as an indicator for bacterial activity in Uranium spheroids in continental rocks.

The general content is informative and maybe of interest for a broad readership. The method presented seems useful, but needs much more detailed discussion and validation including data about the spheroids themselves and the organic content.

We now provide more detailed discussion and data about the spheroids. We have clarified on lines 146–147 and lines 170-172 the nature of the uranium phases present in the spheroids from the two main sample sets, provided a new GC-MS chromatogram (new Supplementary Figure S1) of the hydrocarbons present in the Dingwall spheroids, provided new carbon isotope data from the organic matter in spheroids at Dingwall and Millport (lines 620–626), and added additional citations to studies describing the spheroids in more detail (Kemp et al. 1994; Parnell, 1996).

So far, the general argumentation for a biotic signature is rather weak and in the current state it won't convince the reader by presenting a half-hearted correlation of the uranium isotope fractionation, the uranium concentration with an undefined organic fraction (%loss by heating?!)

This comment necessitates the following clarifications:

1. The uranium isotopic compositions we measure do not represent organic-bound uranium. At Dingwall the uranium is present as botryoids of uraninite and other mineralized phases disseminated in the bituminous cores, not as organic-bound uranium (Parnell and Eakin, 1987). At Budleigh the uranium is present as coffinite; the correlation between the organic % and the uranium concentration here is actually slightly negative. **To clarify the nature of the uranium phases we have added these new details on lines 146–147 and lines 170-172.** The organic content of the spheroids (% reported in Table S2) largely pre-dates spheroid formation, and is essentially irrelevant to the biogenicity of the spheroids. **We have now pointed this out on lines 89–91.**
2. The extremely heavy uranium isotope compositions measured in our samples are among the heaviest ever reported from a natural low-temperature system. The only contextually appropriate mechanism currently known to generate such heavy compositions is bacterial (enzymatic) uranium reduction. This is therefore the best evidence ever presented that reduction spheroids are biogenic.

Especially if the authors assume a biogenic spheroid formation with bacteria involved

in uranium reduction,

We are not assuming this; it is the hypothesis we are testing.

I strongly recommend checking for bacterial traces i.e. biomarker.

We are happy to report the presence of organic biomarkers (see new GC-MS data added to supplementary information as new Figure S2). However, the organic matter pre-dates the formation of the spheroids themselves and is derived from pre-existing sedimentary organic matter, and not primarily the deep biosphere. The composition of the organic material is thus irrelevant to the question whether the spheroids themselves are produced by bacterial reduction — in fact, there is no obvious way to show this *except* by means of studies of the reduced phases themselves, and especially of the uranium minerals that are ubiquitously present. Until recently it was very difficult to evaluate uranium minerals at the low concentrations found in the matrix and some haloes for stable isotopes — hence the novelty, timeliness and significance of our results.

There are numerous geomicrobiological studies describing different bacteria involved in uranium reduction such as iron reducing bacteria and sulphate reducing bacteria (Ontiveros-Valencia et al., 2017; Komlos et al., 2008; Ollmann et al., 2006; Wilkins et al., 2005). Biomarker for these bacteria are well described (e.g. Taylor & Parks 1985; Dowling 1986; Vainsthein et al., 1992; Härtner et al., 2005) and the size of the spheroids would allow a sampling and biomarker investigation (core, halo, adjacent sediment) for microbial lipids. Others describe in detail how and where the uranium could interact/be trapped together with the lipids (Alessi et al., 2014) especially phosphate bearing head groups.

As we have now clarified on line 67, some reduction spheroids contain organic carbon in their cores. As noted by Hofmann (1993), “this organic matter might be directly related to the origin of the reduction spots or it might not be related [...] It seems most likely that this organic material is derived from an inert precursor material.” This means that even clear differences in biomarker content between the inside and the outside of the spheroids would not provide clear supporting evidence for the biogenicity of the spheroids, since these differences are likely to pre-date the spheroids themselves. Moreover, as noted above and **now clarified on lines 146-147 and 170-171,** the uranium isotopic compositions we measure are associated with inorganic phases, not lipids.

A convincing example for biosignatures from a deep continental biosphere .i.e. extreme biogenic isotope fractionation values in combination with complementary techniques was published by Drake et al 2015.

There are of course many convincing examples of biosignatures from the deep continental biosphere, including the excellent paper by Drake et al., which presents organic biomarkers, carbon isotope data from carbonate, and sulphur isotope data from pyrite. However, these conventional biosignatures are unfortunately not relevant to reduction spheroids. Organic matter is commonly present in reduction spheroids but pre-dates them, and pyrite is usually absent, as is carbonate. **We**

have now clarified on lines 86-88 that authigenic pyrite has been found in some reduction spheroids and has an isotopic composition consistent with but not diagnostic of biotic sulphate reduction. In contrast, uranium enrichment appears to be a universal feature of reduction spheroids that could only be explained by in situ uranium reduction in the subsurface. Hence, the measurement of uranium isotopes is uniquely well suited to testing the hypothesis that reduction was biologically mediated.

Find below some titles and excerpts of the recommended publications which may help to address the requested changes.

- Ontiveros-Valencia et al., 2017 Total electron acceptor loading and composition affect hexavalent uranium reduction and microbial community structure in a membrane biofilm reactor.

“SRB Desulfovibrionales, along with sulfur-oxidizers Thiobacteriales, were important in the biofilm during the initial phase that involved SO_4^{2-} reduction; however, only Desulfovibrionales remained in the biofilm that carried out U(VI) reduction. The main URB appeared to be spore-forming Clostridiales and Natranaerobiales, which thrived at low total electron acceptor loadings. Respiration of HCO_3^- by methanogenesis (by Methanobacteriales) was important when the only competing electron acceptor was U(VI), but diminished with addition of BES, the onset of SO_4^{2-} reduction, or major homo-acetogenesis.”

This publication is a study of modern populations, not ancient biosignatures. Unfortunately, it does not describe molecular signatures with relevance to the fossil record.

- Alessi et al., 2014 The product of microbial uranium reduction includes multiple species with U(IV)–phosphate coordination.
“...we have demonstrated unambiguously that noncrystalline U(IV) is comprised of multiple U(IV) species bonded to phosphate groups. Some species are associated with phosphate functional groups in the microbial biomass as either monomers or polymers while others form through the precipitation of inorganic phosphate polymers that are more difficult to extract using a concentrated bicarbonate solution.”

This publication is not relevant to our work since it describes modern populations, not ancient biosignatures.

- Drake et al., 2015 Extreme ^{13}C depletion of carbonates formed during oxidation of biogenic methane in fractured granite.
“...”Here we show that oxidation of biogenic methane in carbon-poor deep groundwater in fractured granitoid rocks has resulted in fracture-wall precipitation of the most extremely ^{13}C depleted carbonates ever reported, $\delta^{13}\text{C}$ down to 125‰ V-PDB. A microbial consortium of sulphate reducers and methane oxidizers has been involved, as revealed by biomarker signatures in the carbonates and S-isotope compositions of co-genetic sulphide.....”

See comment on this paper above.

- Komlos et al., 2008. Long-term dynamics of uranium reduction/reoxidation under low sulfate conditions.
This paper confirms an association between U(VI) reduction and Fe(III)

reduction but is otherwise not relevant to our study since it does not discuss ancient biosignatures.

- Ollmann et al., 2006. Metal binding by bacteria from uranium mining waste piles and its technological applications.
“...Bacillus sphaericus JG-A12 from a uranium mining waste pile in Germany are able to accumulate high amounts of toxic metals such as U, Cu, Pb, Al, and Cd...”
Such metals are redox sensitive; their accumulation is not a reliable biosignature.
- Wilkins et al 2005 The impact of Fe(III) reducing bacteria on uranium mobility
This paper confirms an association between U(VI) reduction and Fe(III) reduction but is otherwise not relevant to our study since it does not discuss preservable biosignatures.
- Härtner et al., 2005. Occurrence of hopanoid lipids in anaerobic Geobacter species
Hopanoids are common biomarkers not specifically associated with Geobacter or with uranium reduction. Organic biomarkers are inappropriate for determining the biogenicity of reduction spheroids.
- Vainshtain et al 1992 Cellular Fatty Acid Composition of Desulfovibrio Species and Its Use in Classification of Sulfate-reducing Bacteria
Organic biomarkers are inappropriate for determining the biogenicity of reduction spheroids.
- Taylor & Parks 1985 Identifying Different Populations of Sulphate-reducing Bacteria within Marine Sediment Systems, Using Fatty Acid Biomarkers.
We truly appreciate the reviewer’s efforts in supplying this detailed list of references and excerpts. However, all of these papers relate to extant microbial communities. To the best of our knowledge, no lipid biomarker uniquely associated with iron- or uranium-reducing microbes has ever been reported from the fossil record. Moreover, as explained above, organic biomarkers are not an appropriate means of investigating the biogenicity of reduction spheroids. **We have clarified in the title and abstract of our manuscript that the reduction spheroids we are investigating are geologically ancient. This study sought evidence of ancient, not recent, bacterial activity.**

Reviewer #2 (Remarks to the Author):

Review: 'A uranium isotope record of the deep continental biosphere' by McMahon et al.

The study by McMahon, v. s. Hood and Parnell is looking at $^{238}\text{U}/^{235}\text{U}$ signatures in a range of reduced spheroids in ancient red beds. The spheroids are believed to have been formed at considerable depth and, if a biotic origin can be confirmed it would provide evidence for bacterial activity deep in the continental crust during Earth's earlier history. The study shows heavier U isotope signatures in U rich centres of spheroids, which strongly suggest a biotic origin for reduced and precipitated U within these haloes. I am not an expert on spheroid and deep Earth biological activity, but I find the study exciting from an U isotope perspective, its importance seems clear to me and the manuscript is well written. It should have a wide audience and I would find it suitable for publication in Nature Communication. Below, I have some comments to specific places in the manuscript that would be good to be addressed by the authors.

We appreciate the reviewer's supportive remarks.

Specific comments

Line 45-46: 'ancient' twice in sentence, consider rewriting

We have replaced the second "ancient" with "pre-modern".

Line 58: 'and' instead of 'are'

We have made this correction.

Line 104-106: I would like to see some word of caution here, as it is clear that you can generate heavier U isotopes through the nuclear field shift effect through abiotic U redox changes. This has been shown in a range of studies focussing on U redox exchange processes (e.g. Bigeleisen 1996, Fujii et al. 2007). It is true that the cited studies have perhaps focussed on abiotic U reduction processes that are closer to 'natural' situations, so there may be an argument for the distinction between abiotic vs biotic reduction based on these, however more work on this is needed before it is clear that abiotic U reduction does not lead to fractionation with heavier U in the reduced phase. Something like 'Experimental studies so far have shown that, by contrast, abiotically ...' or similar wording could cover this caveat.

We agree with the reviewer and **we have adopted the suggested wording on the line in question.** The overarching issue here is the much remarked discrepancy between, on the one hand, theoretical predictions of ^{238}U enrichment in U(IV) verified by chromatography column experiments, and on the other hand, observed natural systems in respect of equilibrium fractionation during abiotic redox reactions. It may be that the nuclear field shift effects under natural conditions "are muted or absent in the kinetics of abiotic reduction reactions" but facilitated by certain bacterial enzymes (Stylo et al. 2015). **We have expanded the supplementary discussion on lines 527–531 to reflect this issue, including citations to the two papers mentioned by the reviewer.**

Line 107-109: Following on from above, I am sceptic about the statement that isotopically heavy U being a ‘powerful proxy’ for ‘their mode of origin’ (abiotic vs. biotic). Making this into a ‘powerful proxy’ will need much more work and simplistic statements misrepresent where the field is in regard to such understanding.

The use of isotopically heavy uranium as a “bio-proxy” has received some high-profile experimental and geological support in recent years (e.g., Stylo et al. 2015, Battacharyya et al. 2017). However, we agree with the reviewer that more work is needed on the uranium “stable” isotope system and that the early stage of the development of the field should be acknowledged. **We have therefore replaced “is a powerful proxy” with “is emerging as a new and potentially powerful proxy” on the line in question.** See also our response to the reviewer’s comment beginning “Line 177-179” below.

The more extended discussion on this in the supplementary material does provide some further information, but it still misses some fundamental important observations from the literature. There are plenty of examples of measurements of U isotopes in nature that are significantly heavier than the crustal average, and is not related to biotic U reduction e.g. zircons and other accessory high temperature phases (Hiess et al. 2012). Also, heavier (and lighter) U isotopes are associated with seawater alteration of mafic ocean crust (Andersen et al. 2015, Noordmann et al. 2016), which is uncertain how this relates to biotic or abiotic processes.

We accept the reviewer’s point that uranium isotopically heavier than the crustal average has been reported from a small number of natural samples in some settings, and **we have revised our supplementary discussion of uranium isotope systematics to reflect this fact, including new citations (lines 540–553).**

That said, in this case of studying these reduction spheroids, the similarity with redox-front hosted U deposits are probably of more significant importance. Redox-front U systems are very likely microbial driven and leads to heavy U isotope signatures in the reduced phase (as discussed earlier in the manuscript and supplementary material). Thus, if similar behaviour is observed in the reduction spheroids, this is likely linked to the same process and therefor it can, in this case, be used to test for if the redox haloes are likely to biotic in origin. This observation should form the basis for the interpretations here and it would be good if the text reflects this better.

We agree with the reviewer’s point that our interpretation our results receives important support from the study of redox front deposits. **We have therefore made the relevance of biogenic heavy uranium phases in roll-front deposits more explicit on line 117-121 and on lines 210–213.**

Line 122-152: I am not an expert on spheroids and from reading these descriptions it is not clear to me if the interpretations of their origin at depth in the basement is based on observations done in this study or if it is this based on literature work e.g. line 126-135 is all or only some of these observations from Parnell & Eakin, 1987? Although the one author from this study is also on the paper, it would be good to make this clearer and if these descriptions are presented for the first time and if the case, some

supplementary material to describe and support these statements this in more detail would seem appropriate.

The reviewer's point is well taken. The inference of formation depth for the Dingwall spheroids is based on descriptions and data that have been published before by one of us (JP), but the inference to a particular depth is novel here; to clarify where the novelty lies in this section, **we have added such phrases as “As previously described by” and “we therefore infer that”, as well as additional citations in this section.**

Also, all these settings are Phanerozoic (apart from one of the additional samples) so it is not expected to be deep biological activity happening?

The question is not whether the deep biosphere existed in the past (it surely did, as noted in the abstract and on line 221-223) but whether it was responsible for the redox processes that formed these reduction spheroids. To address this requires a way to link the reduced phases concentrated in the spheroid with biological activity. This question did not prompt a revision.

Line 176-177: the natural range in $^{238}\text{U}/^{235}\text{U}$ is larger than the 2 permil variability observed here (e.g. see compilation in Hiess et al 2012).

We thank the reviewer for pointing this out. **We corrected our statement that the 2 permil variability was “almost the entire natural range...” by adding “approximating” the qualifier “in low-temperature systems” on the line in question.** The high $^{238}/^{235}\text{U}$ values in the cited compilation are from magmatic crystals and record (probably mantle-derived) high-temperature fractionation processes that are not relevant to the composition of low-temperature authigenic uranium phases on redox fronts. As in the more recent compilation by Andersen et al., 2017, the vast majority of U isotope samples from low-T (sedimentary) systems fall within this range with only some outliers from U ore deposits.

Line 177-179: see comments line 104-109 – ‘strong evidence’, maybe reword.

We agree with the reviewer, and **have revised the wording here to indicate, in agreement with the reviewer's comments, that “Field-based, experimental and geological studies to-date strongly suggest that this pattern of U^{238} enrichment is best explained by bacterial uranium reduction and precipitation within the spheroids” with appropriate citations. The reviewer's comment also prompted a revision to the last sentence of the abstract.**

Line 201-202: two times ‘demonstrate’ in the same sentence consider rewriting.

We have replaced the first “demonstrate” with “evince”.

REVIEWERS' COMMENTS:

Reviewer #1 (Remarks to the Author):

The manuscript entitled "A Uranium Isotope Record of the Ancient Deep Continental Biosphere" reports about Uranium isotope fractionation as an indicator for bacterial activity in Uranium spheroids in continental rocks.

I have reread the paper and the response to my concerns. I think the arguments have been greatly strengthened and the argumentation chain is now much more detailed and clear.

This version of the manuscript can be published in Nature Communications.

Reviewer #2 (Remarks to the Author):

2. review of McMahon et al. 'A uranium isotope record of the deep continental biosphere'

As with the previous version, I enjoyed reading it and think it would provide a strong manuscript for Nature communications. Based on the U isotope systematics the authors have a strong case for their arguments of the U reduction in the spheroids being of a bacterial origin, without simplifying the U systematics too much (as was the case for the previous version but this has been modified in this version). I am no expert on organic biomarkers and I am therefore not in a position to critically evaluate this part of the manuscript. As such I have not much to add apart from some minor comments below regarding typos/wordings and some details in the manuscript.

line 31: not sure that it is correct wording, perhaps, 'U-rich with an heavy isotopic signature' would be more correct

line 56: ... (but not exclusively) ...line 98 remove ':'

line 103: define $\delta^{238}\text{U}$ here first time it is introduced and remove definitions later e.g. line 193

Line 117: add the number quoted is $\delta^{238}\text{U}$ (average $\delta^{238}\text{U}$ of ..)

Line 153: could examples of the other uranium minerals be mentioned, e.g. in brackets after?

Line 202 replace 'uranium isotope composition with ' $\delta^{238}\text{U}$ ' (the former could be either $^{234}\text{U}/^{238}\text{U}$ or $^{235}\text{U}/^{238}\text{U}$)

Line 221: ^{238}U , actually to be technical it could be ^{235}U depletion, so perhaps best to say high $\delta^{238}\text{U}$?

Either in methods or supplementary methods: there are no mentioning of the $^{234}\text{U}/^{238}\text{U}$ ratios, how was these measured?

References: some are repeated (e.g. Lovley et al. 1991 vs. Lovley & Phillips, 1993), look through whole list and correct

Line 572: perhaps something like '... has so far only been observed ..' instead of is only observed

Line 643-647: I miss to notice how very low some of the haloes are in $^{234}\text{U}/^{238}\text{U}$ (0.47)! a detail, but the higher $^{234}\text{U}/^{238}\text{U}$ in the matrix compared to haloes could also be from direct ^{234}U recoil processes, with ^{234}U ejection out of minute uraninite-grains (or similar high U bearing phases)

and into the surrounding matrix with lower U concentrations (similar to what Kigoshi et al. (1971 Science) say for experiments using zircon grains surrounded by a liquid medium). In this case the redistribution of ^{234}U can still be of an ancient origin and at steady-state, despite the disequilibrium and therefore not necessarily a not indicator of recent weathering (<2 Ma).

Point by point

Reviewer 1's comments required no further changes.

Reviewer 2:

As with the previous version, I enjoyed reading it and think it would provide a strong manuscript for Nature communications. Based on the U isotope systematics the authors have a strong case for their arguments of the U reduction in the spheroids being of a bacterial origin, without simplifying the U systematics too much (as was the case for the previous version but this has been modified in this version). I am no expert on organic biomarkers and I am therefore not in a position to critically evaluate this part of the manuscript. As such I have not much to add apart from some minor comments below regarding typos/wordings and some details in the manuscript.

We appreciate these supportive remarks.

line 31: not sure that it is correct wording, perhaps, 'U-rich with an heavy isotopic signature' would be more correct

We have followed this recommendation.

line 103: define d238U here first time it is introduced and remove definitions later e.g. line 193

We have followed this recommendation.

Line 117: add the number quoted is d238U (average d238U of ..)

We have followed this recommendation.

Line 153: could examples of the other uranium minerals be mentioned, e.g. in brackets after?

We have now specified that the uranium is present in xenotime and brannerite as well as uraninite.

Line 202 replace 'uranium isotope composition with 'd238U' (the former could be either 234U/238U or 235U/238U)

The phrasing has been changed to "yielded $\delta^{238}\text{U}$ values near the average crustal value" for clarity.

Line 221: 238U, actually to be technical it could be 235U depletion, so perhaps best to say high d238U ?

We have followed this recommendation.

Either in methods or supplementary methods: there are no mentioning of the 234U/238U ratios, how was these measured?

Table 1 and the existing methods text shows which uranium isotopes were measured and how, so this comment did not prompt a revision.

References: some are repeated (e.g. Lovley et al. 1991 vs. Lovley & Phillips, 1993), look through whole list and correct

We have fixed this error.

Line 572: perhaps something like '... has so far only been observed ...' instead of is only observed

We have followed this recommendation, adopting the wording "has only

been observed (so far)”

Line 643-647: I miss to notice how some of the ~~haloes~~ **cores** are in $^{234}\text{U}/^{238}\text{U}$ (0.47)! a detail, but the higher $^{234}\text{U}/^{238}\text{U}$ in the ~~matrix~~ **halos** compared to ~~haloes~~ **cores** could also be from direct ^{234}U recoil processes, with ^{234}U ejection out of minute otheruraninite-grains (or similar high U bearing phases) and into the surrounding ~~matrix~~ **halos** with lower U concentrations (similar to what Kigoshi et al. (1971 Science) say for experiments using zircon grains surrounded by a liquid medium). In this case the redistribution of ^{234}U can still be of an ancient origin and at steady-state, despite the disequilibrium and therefore not necessarily a not indicator of recent weathering (<2 Ma).

We think the reviewer made a small mix-up in writing this comment; our interpretation of what the reviewer meant (as suggested by the isotopic patterns described here and the quoted value of 0.47) is indicated by my strike-throughs and annotations here. However, the reviewer’s point is interesting and useful. The standard view is that ^{234}U is lost to groundwater but we have added a line to Supplementary Note 2 suggesting the reviewer’s alternative interpretation, which we find plausible.